# Insight into Drug Resistance in Status Epilepticus: Evidence from Animal Models

**DOI:** 10.3390/ijms24032039

**Published:** 2023-01-20

**Authors:** Fei Wang, Qingyang Zhang, Yu Wang, Junzi Chen, Yi Wang

**Affiliations:** 1Key Laboratory of Neuropharmacology and Translational Medicine of Zhejiang Province, School of Pharmaceutical Sciences, Zhejiang Chinese Medical University, Hangzhou 310053, China; 2Department of Neurology, The Third Affiliated Hospital of Zhejiang Chinese Medical University, Hangzhou 310053, China

**Keywords:** status epilepticus, drug target, animal models, hypothesis

## Abstract

Status epilepticus (SE), a condition with abnormally prolonged seizures, is a severe type of epilepsy. At present, SE is not well controlled by clinical treatments. Antiepileptic drugs (AEDs) are the main therapeutic approaches, but they are effective for SE only with a narrow intervening window, and they easily induce resistance. Thus, in this review, we provide an updated summary for an insight into drug-resistant SE, hoping to add to the understanding of the mechanism of refractory SE and the development of active compounds. Firstly, we briefly outline the limitations of current drug treatments for SE by summarizing the extensive experimental literature and clinical data through a search of the PubMed database, and then summarize the common animal models of refractory SE with their advantages and disadvantages. Notably, we also briefly review some of the hypotheses about drug resistance in SE that are well accepted in the field, and furthermore, put forward future perspectives for follow-up research on SE.

## 1. Introduction

Status epilepticus (SE) is traditionally defined as one seizure lasting for more than 30 min or repeated multiple seizures lasting for more than 30 min, with no return to the baseline state of consciousness during the seizure. However, the International League Against Epilepsy (ILAE) redefined SE as a condition resulting either from the failure of the mechanisms responsible for seizure termination or from the initiation of mechanisms, leading to abnormally prolonged seizures (after time point t1, usually 5 min), which causes various long-term outcomes (after time point t2, usually 30 min), including neuronal death, neuronal injury, and the alteration of neuronal networks. These two time points show clear clinical implications: t1 determines the time at which treatment should be considered or started, whereas t2 determines how aggressively treatment should be implemented to prevent long-term consequences [1]. According to available reports, the global annual incidence of SE is 5 to 36 cases per 100,000 adults [2,3,4]. Young children and the elderly are particularly at high risk for SE attacks. The etiology of SE is complex; it can often be due to the inappropriate use of AEDs and the discontinuation of medication without medical advice. Acute encephalopathy, psychiatric factors, overexertion, trauma, tumors, or drug intoxication also induce SE, in addition to a small number of patients whose cause of SE is not yet clear.

Rapid and efficacious treatment is required when patients enter the state of SE. However, current clinical treatments (please see details in Section 2) do not control SE well, leading patients to be likely to develop complications such as brain edema and necrosis, metabolic acidosis, liver function damage, rash, and arrhythmia [5,6]. This situation suggests the significant importance of basic research on the mechanism of SE and further development of effective and safe AEDs.

This retrospective review of the current status of drug therapy for SE summarizes the animal model of refractory SE and the hypothesis for drug resistance, and puts forward prospects for SE. We systematically searched PubMed for publications in English with the combined keywords, including “Status Epilepticus”, “animal models”, “hypothesis”, “drug resistance”, “refractory”, “antiepileptic drugs”, and “drug target”. This generated a total of 247 articles and we selected papers published after 2000. Reference lists of relevant papers were also checked for additional studies. We also used a few earlier classic articles from before 2000 if they were particularly pertinent to the discussion. We arbitrarily chose seminal work, clinical studies with the highest level of evidence, and some articles were excluded from this review due to missing details. Finally, 152 publications were included in the present review.

## 2. Current Dilemmas in Pharmacological Treatment

Drug therapy is currently considered the most common clinical treatment and, in most cases, only intravenous agents are used for SE. Benzodiazepines are the first choice treatment, and mostly consist of lorazepam or diazepam that is adopted as an initial monotherapy treatment for SE. Real-time monitoring of vital signs is maintained while the patient is in the observation phase. When SE attacks for 5 min, commonly used therapeutic agents are: diazepam, lorazepam, and clonazepam [7,8]. In recent years, the use of lorazepam instead of diazepam has been extensively recommended, but there is currently no lorazepam in some countries, including China, making diazepam-based treatment still the first choice in these countries. However, given that the action intensity of clonazepam is 10 times that of diazepam, clonazepam is frequently considered a prior option for SE treatment, although more clinical experience is needed for it to earn widespread recognition. There is also a narcotic drug, midazolam. Although many studies now find it to be more effective than diazepam, it also has stronger effects on respiration and blood pressure than diazepam, and there are limitations to its use as an anesthetic; this drug is not yet widely recognized, so we have not placed it among the first-line drugs in this article [9].

When SE lasts for more than 5 min and less than 30 min, some non-sedative AEDs, such as intravenous formulation of phenytoin, valproic acid, or levetiracetam, are used as second-line medications [10,11]. In clinical studies, these three drugs, when used alone, have a nearly 50% chance of stopping a seizure within 60 min [11,12]. Phenytoin is more effective than levetiracetam in convulsive SE of children, but levetiracetam may also be a good alternative for safety and patient adaptation [13,14]. At this stage, drugs can be administered in combination. If the patient’s SE fails to terminate within this time, there is a high probability that it will evolve into refractory SE (RSE), i.e., a condition that occurs when two or more AEDs including at least one non-benzodiazepine drug are not effective [15].

When SE lasts longer than 30 min, an RSE situation occurs, and anesthetics, including propofol, ketamine, and midazolam, can be further used for the treatment of RSE and are widely accepted [16]. However, super-refractory SE (SRSE) occurs when SE lasts for at least 24 h after the initiation of continuous anesthetics (i.e., midazolam, propofol, pentobarbital, and ketamine) or during the weaning of these drugs [17,18]. Patients entering the stage of SRSE can choose a ketogenic diet, hypothermia, and electroconvulsive therapy [19,20,21]. At the same time, factors including the airway, blood pressure, temperature, intravenous electrocardiography, complete blood count, glucose, electrolytes, arterial blood gas, and tox screen should also be tested. Considering that each drug has certain side effects of lowering blood pressure and respiratory depression, much attention should be paid to the patient’s condition in the case of giving these drugs [11,22]. Thus, the early initiation of anticonvulsants is the key to the successful control of SE. Clinical data indicate that a longer duration of SE generally implies a more dangerous prognosis [23,24]. Thus, the principle of treatment for SE is: “time is brain” (Figure 1).

At present, although there are many AEDs available to be chosen, the following dilemmas still need to be faced for contemporary drug treatment: (1) The therapeutic window is short as mentioned before; and delayed drug therapy is prone to inducing drug resistance and long-term adverse consequences [25,26]. (2) SE is prone to recurrence after drug treatment. In recent years, some studies have found that the recurrence rate of SE in adult patients is about 20% over the next 4 years [27,28]. (3) SE remains at risk of misdiagnosis [29,30]. Non-epileptic seizures can be misdiagnosed as genuine epileptic seizures, which can lead to inappropriate, costly, and potentially harmful treatment. SE is a life-threatening condition, so early determination is very important.

Thus, what has caused this situation and how can these dilemmas be addressed? On the one hand, the current source of AEDs shows that most drugs are screened through acute animal models [31]. As epilepsy is a chronic disease, there is a great lack of screening candidate compounds in drug-resistant chronic epilepsy models or drug-resistant SE models. On the other hand, most of the current AEDs produce antiseizure actions with the mechanism of “excitation-inhibition imbalance”, and there is a lack of new drug targets and new mechanisms for initiation, so it is crucial to clarify the mechanism of action of refractory SE and develop effective compounds.

## 3. Animal Models of SE

Animal models are of great significance for investigating the pathogenesis of SE and the drug resistance mechanism, and are even more important today. Each model has different characteristics, and the current animal SE models used in studies are characterized by prolonged seizures or recurrent multiple seizures. Based on extensive literature research, we selected several widely used and easily manipulated modeling approaches, including kainic acid (KA), pilocarpine, kindling, and prolonged febrile seizure (FS) models. We discuss each model in the details below (Table 1). In the present review, we focus only on the SE phase, especially studies investigating the termination of SE.

### 3.1. KA-Induced SE Model

KA, an L-glutamate analog, is an effective stimulant in the central nervous system (CNS) with activation of glutamatergic transmission, which selectively binds ion-shifted glutamate receptors (KA receptor). Irregular high-amplitude spikes first appear after KA administration in the brain. Then, periodic epileptiform discharges, high-frequency bursting, and a combination of periodic epileptiform discharges with increasing severity of epileptiform discharges as well as high-frequency bursts of short duration appear gradually [32].

KA is usually administered in three ways, including systemic, intracerebral, and intranasal injection [33]. When animals are intraperitoneally injected with KA, behavioral seizures occur within approximately 15–30 min, followed by SE at 30–90 min. SE can last 2–6 h, and neurons begin to be damaged 3 h after KA injection. The mortality rate of this model is about 47–75% [34,35,36,37]. The advantage of intraperitoneal KA injection is that the method is simple and easy to operate, which eliminates the confounding effects of surgical anesthesia and the additional damage caused by direct contact with brain tissues during the intracerebral injection. However, it is not easy to control the bioavailability of KA in the brain and high mortality. For intracerebral injection, the animals are first implanted with a cannula, and KA is injected into the hippocampus or amygdala, or lateral ventricles through the cannula. This approach is endowed with the advantage that KA can bypass the blood–brain barrier (BBB) and go directly to the designated site, thereby reducing mortality. Usually, the animal reaches SE within 30 min and the SE lasts for 3–12 h. The mortality rate of this model is about 8–21% [32,38,39]. The disadvantage is the operation inconvenience and the requirement of well-trained researchers to establish this model. Intranasal delivery is relatively uncommon. By modeling in this way, the animals reach SE within 15–30 min and this lasts for 1–5 h [40,41]. The mortality is lower than that of intrahippocampal administration, which is a nice advantage. However, similar to systemic delivery, it fails to easily control the bioavailability of KA.

In the KA-induced SE model, diazepam with delayed treatment easily induces the drug-resistant state. Some studies have found that in the very early stage of KA-induced SE (about 5 min), 50 mg/10 mL of diazepam eliminated the epileptiform electroencephalogram (EEG) activity and behavioral seizure activity was also terminated. However, the EEG pattern of SE recurred in the diazepam-treated animal at 1 h after treatment, which was accompanied by subtle behavioral seizures [42]. Our group previously found that the diazepam, delivered immediately (10 min) after the onset of SE, exerted a termination effect on SE for more than 60% efficacy and reduced the EEG power of SE. However, after 40 min of SE, diazepam treatment cannot control SE and animals often develop drug resistance [43,44]. This strongly suggests that the KA-induced SE model combined with delayed treatment of diazepam might be used as a model of drug-resistant SE.

### 3.2. Pilocarpine-Induced SE Model

Pilocarpine is an alkaloid extract from the leaves of the genus rutaceous, which mainly acts on muscarinic receptors and shows a muscarinic-like effect. The nicotinoid effect can also be observed in large doses. It initiates seizures by activating cholinergic receptors and N-methyl-D-aspartic acid (NMDA) receptors to maintain SE [45,46]. Pilocarpine models can be combined with other drugs, the most widely used is the lithium–pilocarpine combination.

After intraperitoneal administration of pilocarpine, the animals develop discontinuous seizures after about 30 min, which are followed by SE within 1–2 h, lasting for about 90–150 min. However, the mortality rate in this model is high, about 27.4–40%. The mortality rate can even reach 85% at a high dosage of pilocarpine [45,47,48]. SE can also be induced by intracerebral administration of pilocarpine, including direct injection into the dorsal hippocampus or ventricles [49,50]. Spikes appear approximately 25–30 min after administration and have a high peak rate. The epileptiform discharges consist of rapid spikes, which then change to rapidly repeating bursts and decrease in frequency over the next few hours. Such SE can last for at least 5 h [51,52].

Several studies have found good termination effects with diazepam injected 10 min after pilocarpine injection; however, the longer the time window of the treatment, the more dose-dependent the effects of diazepam become. Within 45 min after pilocarpine injection, animals enter a state of SE, with a reduced response to diazepam and resistance. However, complete recovery from SE can be achieved after high doses of diazepam (100 mg/kg), with dose dependence occurring 30 min after pilocarpine injection [53,54]. Together, both KA and pilocarpine-induced SE models reflect a clinical phenomenon that predisposes SE to a resistant state under conditions of delayed pharmacological intervention.

### 3.3. Kindling-Induced SE Model

Kindling refers to the repeated electrical stimulation of limbic structures, which is a widely used model of temporal lobe epilepsy [55,56]. The most commonly stimulating site of kindling is the hippocampus or amygdala. Kindling has also been used as an electrically-induced SE model in some cases.

Intermittent electrical stimulation of the hippocampus lasting 90 min induces SE [57]. SE induced by electrical stimulation of the hippocampus initially consists of a rapid spiking pattern of electrical spasms, after which the frequency of the spike decreases over the next hours. Then the EEG switches to rapid, repetitive bursts. The SE ends when these bursts switch to periodic discharges. After kindling stimulation is terminated, the animal exhibits persistent wet-dog-like behavior [57]. Sustained electrical stimulation in the basolateral amygdala lasting for 25 min induced three different types of SE: Type 1, partial (non-convulsive) SE, which was characterized by limbic stage 1 and stage 2 seizures (Racine Scale); Type 2, partial SE interrupted by occasional periods of stage 3 and generalized convulsive (stage 4 or 5) seizures; and Type 3, generalized convulsive SE. During SE, hippocampal extracellular acetylcholine (ACh) levels were significantly increased, but the extracellular ACh precursor choline levels were not significantly increased. At the same time, the levels of extracellular γ-aminobutyric acid (GABA) and glutamate in the hippocampus increased significantly during SE, and glutamate levels were further increased after the termination of Type 3 SE [58]. No detailed report has been found on the EEG of SE induced by electrical stimulation of the amygdala.

There are few reports on drug resistance in the kindling-induced SE model. Available studies have shown that SE can be terminated in 65% of animals when phenobarbital is injected within 10 min after SE. It can be terminated in 100% at the second injection. In addition, it has been reported that phenobarbital injected 15 min after the onset of SE can be controlled in more than half of the animals. Only 33% of the animals could be controlled when phenobarbital was injected 30 min after the onset of SE, and this percentage was even lower when phenobarbital was injected more than 60 min after the onset of SEI [59]. This suggests that the drug does not effectively intervene in the time window of kindling-induced SE, even after prolongation.

### 3.4. Prolonged FS-Induced SE Model

Clinically, FS often occurs in children between 3 months and 5 years of age [60,61,62], affecting 2–5% of children worldwide. The ILAE defines an FS as a seizure occurring in childhood after one month of age, associated with a febrile illness that is not caused by an infection of the CNS. A child with the diagnosis of FS cannot have a history of neonatal seizures, a previous unprovoked seizure, or meet the criteria for other acute symptomatic seizures [60]. FS can be divided into simple and complex types. Simple FS usually occurs only once and lasts less than 15 min, most children recover from these within a few minutes and have a good prognosis. Complex FSs are focal attacks that generally occur multiple times during periods of fever and last more than 15 min. Prolonged FS will lead to febrile SE (FSE) [63]. As a subtype of complex FS, FSE is a neuroemergency, leading to long-term complications and predisposing patients to other neurological disorders in later life [64,65]. Therefore, designing an effective animal model of prolonged FS is of great significance to study the pathogenesis changes and therapeutic interventions of FSE.

Three types of methods have been commonly used to model FSE: (1) Lipopolysaccharide (LPS)-induced FS. Usually, LPS (10 mg/kg) was given to the animals by intraperitoneal injection for 2 h before inducing FS [66]. (2) Heat-induced FS. The mice were placed in a constant high-temperature chamber to induce FS, and the induction was repeated three times [67]. Alternatively, mice were given a hot water bath every two days, 10 times for 5 min each time [68], or with heated lamps [69]. (3) FS induced by heat combined with LPS. For example, mice were injected intraperitoneally with LPS and then heated with a hair dryer to induce FS [70].

The EEG features of FSE are not well documented. In cortical recordings, only semirhythmic or sporadic spikes were seen. During normothermia, both amygdala and cortical recordings typically reveal non-rhythmic, low-voltage discharges. When animals enter into FS, rhythmic discharges with increased amplitude occur in the amygdala. At this point, there’s no change in activity in the cortex, but there is a typical decrease in discharge abundance and voltage [71,72].

At present, the drug resistance of FSE has not been reported in detail in animal models. We previously showed that diazepam is effective on the first FS, but cannot effectively control repeated or prolonged FS [73,74,75]. Further study is still needed in the future to investigate drug resistance states in FSE models.

### 3.5. Other SE Models

In addition to the models mentioned above, there are other SE models. As cortical insults are very common triggers of human SE that are difficult to treat, experimental animal studies have often targeted temporal lobe structures to induce SE [76]. It has previously been reported that focal seizures are caused by the implantation of cobalt into the motor cortex [77,78]. Cobalt binds to oxygen, causing functional hypoxia, and low oxygen, the cortical tissue damage caused by trauma, hemorrhage, hypoxia, and infection often leads to acute seizures and SE [79,80]. The animals are then injected with homocysteine, an agonist for the ionized glutamate receptor for NMDA. Activation of NMDA receptors allows Ca^2+^ entry into neurons and enhances excitability, thus triggering prolonged seizures. However, this model can lead to edema and BBB damage, which increases mortality. Phenobarbital and lorazepam have shown therapeutic effects, but no findings have been reported about drug resistance in this model [81].

Organophosphorus chemicals, including pesticides (parathion et al.) and nerve gases (diisopropyl fluorophosphate (DFP), sarin and soman, etc.), have also been used to induce the SE model [82]. These agents potently inhibit acetylcholinesterase, leading to the accumulation of ACh and stimulating the cholinergic system, which causes an acute cholinergic crisis. Seizures can be evident within minutes of organophosphorus exposure and often progress to SE [83]. High doses of DFP have generally been chosen to induce SE in laboratory animals [84,85,86]. DFP has a very short latency to induce SE and can reach SE within 10–20 min [87,88]. In this SE model, phenobarbital was found to terminate SE dose-dependently. Phenobarbital at a high dose (100 mg/kg) was able to completely terminate SE, but it showed severe adverse effects and high mortality rates [87].

In addition, there are other SE models, including the Tustin model [89], thiocolchicoside model [90], and so on. However, the drug resistance situation in these SE models is not clear. The usage rate of these models may not be common due to the difficulty of modeling and the low universality. Although there are many types of SE models, as we mentioned above, the real drug-resistant SE model is limited, which retards our understanding of the mechanism of drug-resistant SE. In other aspects, current SE models are often used to study the later process of epileptogenesis followed by SE. We here emphasize the importance of studying the phase of SE termination, which may be important for developing effective drugs for controlling drug-resistant SE.

## 4. Common Hypotheses of Drug-Resistant SE

In animal studies, there are many hypotheses regarding the drug resistance mechanism of SE, including the classical “excitatory-inhibitory imbalance”, GABAergic transmission, glutamatergic transmission, and ion channels. In recent years, it has been found that neuroinflammation, nutritional factors, and other factors are also significantly involved. We have selected some hypotheses that have gained a high degree of acceptance in the field in recent years (Figure 2).

### 4.1. Degraded GABAergic Transmission Hypothesis

The primary inhibitory neurotransmitter in the brain is GABA [91]. GABA is produced from glutamate, the primary excitatory neurotransmitter, by glutamate decarboxylase with pyridoxal phosphate as a cofactor [92]. The primary function of GABA is to reduce neuronal excitability by binding to various GABAergic receptors in the plasma membrane of both presynaptic and postsynaptic neuronal processes, thereby inducing hyperpolarization [93].

The primary mechanism of benzodiazepines is to increase GABA function, but as SE continues, benzodiazepines become ineffective. It is primarily due to the internalization of synaptic (gamma-subunit)-containing GABA_A_ receptors [54,94]. During SE, both the GABA_A_ receptor-mediated inhibition of hippocampal principal neurons and the response of these neurons to benzodiazepines are reduced [64,65,66]. According to Goodkin et al.’s report, during SE, increased internalization rates of GABA_A_ receptors are modulated by neuronal activity, which contributes to the reduced inhibitory transmission observed during prolonged SE [94]. Another study found that during pilocarpine-induced SE, colocalization of the β2/β3 and γ2 subunits of GABA_A_ receptors with the presynaptic marker synaptophysin is reduced. In contrast, the proportion of these subunits is increased in dentate granule cells [95]. These results suggest two possible explanations for the decrease in GABA_A_ receptors during SE. In one instance, the internalization of ligand-dependent GABA_A_ receptors increases as the extracellular GABA concentration increases. The other is that the increase in GABA_A_ receptors’ internalization is the result of a non-ligand-independent mechanism that is mediated by the excitability of neurons, which is increased by the stimulation of excitatory amino acid receptors [96,97,98]. In addition, the resistance of GABAergic drugs in SE may be due to the phosphorylation status of the potassium chloride transporter KCC2, which increases intracellular chloride levels and decreases the inhibition of GABA_A_ receptor activation [99].

### 4.2. Augment of AMPA Receptor Function Hypothesis

AMPA (α-Amino-3-hydroxy-5-methyl-4-isoxazolepropionic acid) receptors are heterotetramers composed of combinations of the protein subunits GluA1, GluA2, GluA3, and GluA4, which are assembled into ion channels with distinct physiological properties that mediate rapid (millisecond timescale) excitatory neurotransmission [100,101]. When glutamate binds, it activates the AMPA receptor, producing an excitatory postsynaptic potential [102]. AMPA receptors are insensitive to membrane potential, and their opening is gated by glutamate [103].

During SE, pyramidal neurons of the hippocampus are highly active and both AMPAR-mediated transmission and NMDA receptors are activated [104,105]. Seizures enhance AMPAR-mediated transmission in activated neurons by inserting GluA1 into glutamate synapses [106]. Each seizure alters the transmission of a subset of neurons, so recurrent seizures reorganize a larger subset of neurons [106]. These neurons finally form a self-perpetuating and propagating network that ultimately supports SE. Previous studies have shown that AMPA receptor antagonists have a broad spectrum of anticonvulsant activity and terminate SE [102,107,108], even in benzodiazepines-resistant conditions [102]. NS1209, a novel AMPA receptor antagonist, terminated electrically induced and KA-induced SE without relapse after 24 h [109,110]. GYKI52466, another AMPA antagonist, has also been shown to terminate benzodiazepines resistant SE by blocking AMPA receptors in a non-competitive manner via an allosteric site on the receptor channel complex [111]. It indicates that AMPA receptor antagonists may effectively treat benzodiazepines-resistant SE.

### 4.3. Overactivation of Neuroinflammation Hypothesis

Neuroinflammation is a CNS response to various injuries, including tissue damage, infection, autoimmune disorders, stress, and seizures [112]. Neurogenic inflammation is the inflammatory response in the CNS caused by increased neuronal activity in the absence of obvious pathological conditions. Experiments have demonstrated that the overexcitation of neural networks during epileptic seizures results in neurogenic inflammation [95,113]. Activated microglia and astrocytes, neurons, and endothelial cells of the BBB all cause neuroinflammation [114]. It has been shown that the expression of inflammatory mediators, such as Interleukin-1β (IL-1β), high mobility group box 1 (HMGB1), and many other cytokines, is significantly upregulated during SE [43,44,115]. Neuroinflammation is not only a consequence of seizures but also contributes to their genesis. Next, we emphasize the role of IL-1β and HMGB1 in SE by providing the examples listed below:

During recurrent seizures, endogenous “danger signals”, which are activated by potentially pathogenic damage, activate toll-like receptor 4 (TLR4) and brain cells release HMGB1, while IL-1β, released from neurons, glial cells, and brain endothelial cells during inflammasome activation, and macrophages expelled from the blood, activate IL-1R1 in epilepsy [112,116,117]. Similar intracellular signaling molecules are activated by the TLR4 and IL-1R1 receptors. This signaling pathway can be activated by the transcription of inflammatory genes regulated by transcription factors nuclear factor kappa-B (NF-κB) and activator protein 1 or by posttranslational modification of neuronal channel proteins or receptors by other kinases. Signaling through these receptors contributes to neuronal hyperexcitability, leading to seizures and drug-resistant epilepsy [118,119,120,121]. Thus, significant drivers of epileptic neuroinflammation, HMGB1 and IL-1R, are widely regarded as anti-inflammatory therapy targets for treating drug-resistant SE [122,123,124].

Based on the upregulation of HMGB1 expression during SE, we identified the HMGB1-TLR4 pathway as a contributor to diazepam-resistant SE. When the onset of SE was 40 min, diazepam alone was ineffective in terminating SE. Anti-HMGB1 mAb neutralized excessive HMGB1 or knocked down downstream TLR4 receptors, which significantly increased the termination percentage of diazepam-resistant SE, indicating that HMGB1 is a crucial factor in maintaining diazepam resistance. When the onset of SE is prolonged (>60 min), synergistic therapy appears to be more efficacious [43]. The study demonstrated that anti-HMGB1 mAb controls led diazepam-resistant SE with a broad intervention window. On the other hand, the IL-1β protein level increased significantly during prolonged SE, especially in the diazepam resistance condition of approximately 40 min. We found that prolonged SE could be terminated by using an antagonist of IL-1R1, in conjunction with diazepam, and that prolonged SE in IL-1R1 KO mice was not resistant to diazepam. Importantly, we found that IL-1β administration is sufficient to induce diazepam resistance at early-stage SE, implying that accumulated IL-1β may promote the progression of diazepam-resistant SE. The knockdown of the IL-1R1 gene may be able to reverse or prevent the progression of diazepam resistance [44]. Therefore, IL-1R1 is a practical drug target for treating drug-resistant SE.

### 4.4. Upregulated P-glycoprotein (P-gp) Hypothesis

P-glycoprotein (P-gp) is one of the most crucial efflux transporters of exogenous substances of BBB, highly expressed in secretory epithelial cells of various peripheral tissues, while expressing a relatively low level in BBB endothelial cells [125,126]. The efflux effect mediated by P-gp may restrict AEDs’ concentration in the brain [127].

During SE, there is a substantial increase in P-gp expression, which is related to numerous molecular signals. In the pilocarpine-induced SE model, it is reported that HMGB1 knockdown reduces the expression of MDR1A/B mRNA and P-gp protein via the RAGE/NF-κB inflammatory signaling pathway [128]. Deng et al. showed that the overexpression of exogenous miR-146a-5p significantly inhibits the expression of IL-1R-associated kinase and TNF receptor associated factor 6 in the brains of drug-resistant SE rats via the NF-κB signaling pathway, which reduces the level of P-gp, and may represent a promising therapy for drug-resistant SE [129]. The increased expression of P-gp can also be induced by the activation of cyclooxygenase-2 (COX-2). In other words, the increased glutamate levels in the brain due to SE attack activates COX-2 to increase the expression of P-gp. In the KA-induced SE model, Ciceri found that low-dose cefoxib with anti-inflammatory activity could inhibit central COX-2 and P-gp expression [130]. These findings imply that anti-neuroinflammatory drugs with the potential to inhibit central COX-2 may restore the increased P-gp expression, thereby enhancing the distribution of AEDs in the brain and aiding in improving the control of drug-resistant SE.

### 4.5. Neurotrophic Factors (NTFs) Hypothesis

NTFs are molecules that support the growth, survival, and differentiation of developing neurons. In the CNS, they play a functional role at the synaptic level, and exert distinct regulatory effects on excitatory and inhibitory synapses in addition to their neurotrophic effects. The study also suggests that NTFs may play a role in certain aspects of SE [110], including glial-cell-derived neurotrophic factor [131,132], nerve growth factor [133,134], and vascular endothelial growth factor [135]. However, the factors fibroblast growth factor 2 (FGF2) and brain-derived neurotrophic factor (BDNF) receive the most attention.

Previous research indicates that FGF2 is closely associated with seizures. However, it does not appear to be a necessary condition for SE onset, though it may exacerbate SE [136]. Chris Gall and colleagues demonstrated that SE induces dramatic, multi-day increases in the expression of BDNF [137]. This increase correlates with the time course of the SE-induced activation of TrkB signaling [138]. Different downstream pathways may mediate the detrimental and beneficial effects of SE-induced TrkB signaling in SE, and selective inhibition of the detrimental pathway may become a novel therapeutic strategy. For example, activation of chicken phospholipase Cγ1 signaling appears to mediate the detrimental epileptogenic effects of TrkB activation [139]. Based on this signaling pathway, some researchers used CEP-701 to inhibit TrkB activation and found that it can reduce the severity of KA-induced seizures [140]. Currently, NTFs are studied primarily in the final phase of SE. Therefore, targeting the signaling of distinct NTFs for drug-resistant SE remains unclear.

## 5. Summary and Outlook

As we summarized, there are currently many dilemmas in the pharmacological treatment of SE. Despite our review of the common hypotheses of drug resistance mechanisms for SE, current drugs are single mechanism, which is a daunting challenge for multi-causal SE. Many new safe and effective drugs are urgently needed to be designed and developed. Meanwhile, with the advancement of technology and medical treatment, some of the historical data cannot be updated in real time, so we can only select the content that is currently well recognized in the field, including models and hypotheses, which might be the limitation of this article. However, in this review, we provide an updated summary for an insight into drug-resistant SE, hoping that it will help in the understanding of the mechanism of refractory SE and provide for an increase in attention in order to develop active compounds for both research and the industry field. Here we put forward further future perspectives for the follow-up research on SE.

(1)New Mechanism of SE in both molecular and circuit levels

Apart from the pharmacoresistant hypotheses of SE mentioned before, new mechanisms and other drug targets are still needed. For example, miR-15a-5p that targets negatively regulated NR2B expression [141] and the ATP-gated purinergic P2X7 receptor [142] that may function as novel drug targets for not only epilepsy but also SE. P2X7 receptor antagonists can also play an adjuvant role in the SE model of benzodiazepine drug failure. Engel et al. found that lorazepam or P2X7 receptor antagonist A438079 alone had no good effect on drug resistance in long-duration SE, while the combination of the two could effectively terminate SE [143]. In addition to some new molecules, we should pay more attention to the neural circuit mechanism of SE. Epilepsy is gradually being accepted as a circuit-level syndrome pathologically characterized by hypersynchronous seizure activity. At present, although we have elucidated many neural circuitry mechanisms in epilepsy with the development of optogenetics, chemogenetics, in vivo imaging, etc. [144,145], the neural circuit involved in different phases of SE is still largely unknown. A seizure has its own electrographic evolution feature: a seizure initiation, a spread, and a termination, which is usually linked with different networks [146]. As prolonged SE is the failure of the mechanisms responsible for seizure termination or from the initiation of mechanisms, revealing the circuit that medicates the termination of SE will be very important for SE treatment by using the precise intervention of the circuitry approaches.

(2)New smart therapeutics with safe and effective features

SE is a complex process involving complex mechanisms. The hypothetical mechanisms we have mentioned above may be a process of mutual integration between them, and they may all participate in the pathogenesis of SE. Whether monotherapy or a multi-drug combination should be used for optimal treatment (both safe and effective features) of refractory SE is still in debate and needs to be further clarified, especially in double-blind randomized control multi-center research. At the same time, as drug resistance in SE is usually caused by delayed drug treatment, finding more effective and timely treatment strategies, especially drugs that can broaden the time window, is crucial for clinical patients. Previously, we developed “smart” electro-responsive drug carriers [147,148,149], which release drugs quickly in response to epileptiform discharges. This would be a great advantage for timely “on-demand” drug delivery to improve the efficacy of pharmacotherapy for SE.

(3)Prediction of drug resistance in SE

In the future, how to predict drug resistance in advance in patients with SE is also an important direction to control SE. If so, we may be able to prepare and intervene in advance before a patient develops SE. Although some studies are trying to find some biomarkers [150], such as the neuron-specific enolase and progranulin, glia-specific HMGB1 et al., the correct rate of prediction has not been successfully verified in a larger population. Combined with big data analytics, artificial intelligence [151,152], and a deeper insight into the mechanism, we may prospectively identify the risk factors that are apparent for SE or even drug resistance in SE, which may bring the intervention window forward to a much earlier point of treatment.

## Figures and Tables

**Figure 1 ijms-24-02039-f001:**
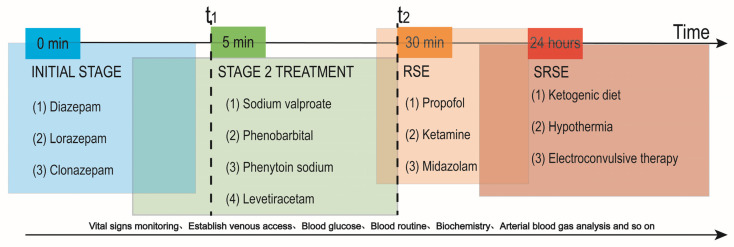
Clinical use of medication at different times during SE attacks. (1) When SE = 0 min, the preferred drugs are diazepam, lorazepam, clonazepam, etc. (2) When 5 min ≤ SE < 30 min, drugs such as phenytoin, valproic acid, or levetiracetam are used as second-line medications. (3) When SE > 30 min, RSE occurs and anesthetics, including propofol, ketamine, and midazolam, can be administered. (4) When SE lasts for 24 h, patients can choose between a ketogenic diet, hypothermia, or electroconvulsive therapy.

**Figure 2 ijms-24-02039-f002:**
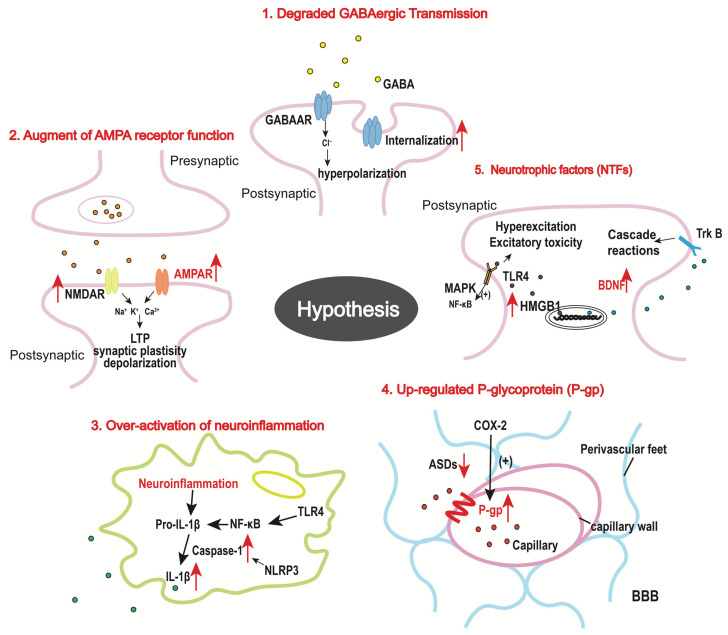
Schematic diagram of main drug resistance mechanism in SE. The red up and down arrows represent the increase or decrease.

**Table 1 ijms-24-02039-t001:** Common animal models of SE.

Model	Mode of Operation	Mechanism	Mortality Rate
KA model	Intraperitoneal injection	KA binds directly to non-NMDA (KA) receptors in the neuronal postsynaptic membrane, producing excitatory postsynaptic potentials that lead to seizures.	47–75%
	Intraventricular injection		8–21%
	Intranasal injection		Lower than intraventricular injection
Pilocarpine model	Lithium–pilocarpine	Pilocarpine can stimulate not only the M receptor, but also NMDA receptors, metabolic glutamate receptors, resulting in activation of the excitatory system in the brain.	27.4–40%
	Intracerebral administration		
Kindling model	Hippocampus	Repeated electrical stimulations cause a gradual change in the excitatory synaptic plasticity and lower seizure threshold.	——
	Amygdala		
Prolonged FS	LPS-induced FS	An imbalance between the excitatory neurotransmitter glutamate and the inhibitory neurotransmitter GABA.	About 50%
	Heat-induced FS		
	FS induced by heat combined with LPS		

## Data Availability

Data sharing not applicable.

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
