# Peer review of "Insight into Drug Resistance in Status Epilepticus: Evidence from Animal Models"

_ijms, 2023, doi:10.3390/ijms24032039_

Round 1

Reviewer 1 Report (Previous Reviewer 2)

the manuscript has been significantly improved so I suggest to publish

Author Response

Reviewer 1:

the manuscript has been significantly improved so I suggest to publish

Reply: Thank you very much for your time involved in reviewing the manuscript and the positive comments on this review.

Reviewer 2 Report (New Reviewer)

Author Response

Reviewer 2:

Firstly, I would like to congratulate the authors on their interesting work. However, there are some points that need to be addressed:

Reply: Thank you very much for your time involved in reviewing the manuscript and the valuable comments on this review. We have revised our paper according to your comments, which really improved the quality of our paper. The following are our responses in a point-by-point manner.

1)  Title: The authors should add the type of study to the title.

Reply: Thanks. We have added the type of study to the title.

2)  Abstract: The authors should briefly describe the methods in the abstract and try to answer questions such as the main research goals? Which are the main research findings? Authors should also remove the term “systematically” as this is not a systematic review.

Reply:

Thank you for these constructive comments. Accordingly, we have described the searching methods, added the research goals and main findings in the revised abstract of manuscript as following:

Revised Abstract:

“Status epilepticus (SE), a condition with abnormally prolonged seizures, is a severe type of epilepsy. At present, SE is not well controlled by clinical treatments. Antiepileptic drugs (AEDs) are the main therapeutic approaches, but they are effective for SE only with a narrow intervening window and easily induce resistance. Thus, in this review, we provide an updated summary for the insight into drug-resistance SE, hoping to be helpful for the understanding the mechanism of refractory SE and developing active compounds. Firstly, we briefly outline the limitations of current drug treatments for SE by summarizing the extensive experimental literature and clinical data through searching the PubMed database, and then summarize the common animal models of refractory SE with their advantages and disadvantages. Notably, we also briefly review some of the hypotheses about drug resistance in SE that are well accepted in the field, and further put forward future perspectives for the follow-up research on SE.”

3)  L-10: I advise authors to remove this definition and instead add in a summarized way ILAE's definition of SE: "Status epilepticus is a condition resulting either from the failure of the mechanisms responsible for seizure termination or from the initiation of mechanisms, which lead to abnormally, prolonged seizures. " Please see this for more information: https://www.ilae.org/guidelines/definition-and-classification/definition-and-classification- archive

Reply: Thank you for your comment. We have revised the definition of SE in a summarized way as following:

“Status epilepticus (SE), a condition with abnormally prolonged seizures, is a severe type of epilepsy.”

4)  L-40: I advise the authors to change this terminology in the text to Antiepileptic Drugs (AED) as this is the latest used by ILAE.

Reply: Thank you for your comments. As suggested by the reviewer, we have corrected the “ASD” into “AED”

5)  L44-50: I advise the authors to work with English editing services to improve the readability of the text. also, work on further describing the type of study and specific goals of this research.

Reply: Thank you for this suggestion. An English native have helped us to polish the language in the revised manuscript. The red part that has been revised according to your comments.

“Rapid and efficacious treatment is required when patients enter the state of SE. However, current clinical treatments (Please see details in section 2) do not control SE well, leading patients likely to develop complications such as brain edema and necrosis, metabolic acidosis, liver function damage, rash, and arrhythmia [5,6]. This situation highly suggests the importance of basic research on the mechanism of SE and further development of effective and safe AEDs.”

6)  L-45: The authors should further clarify this statement and add references to their affirmations. Which clinical treatments?

Reply: Thank you for your comments. For the detailed explanation of this issue, I have mainly placed this section in the "2. Current Dilemmas in Pharmacological Treatment" section. In addition, based on your suggestion, I have revised sentence as following:

“Rapid and efficacious treatment is required when patients enter the state of SE. However, current clinical treatments (Please see details in section 2) do not control SE well, leading patients likely to develop complications such as brain edema and necrosis, metabolic acidosis, liver function damage, rash, and arrhythmia.”

7)  L54-64: The authors should further clarify this statement and add references to their affirmations. Which clinical treatments? This is only one paragraph where the authors have not added appropriate references to their statement but many other references are missing throughout the text and I advise the authors to revise that. (L95-98, and L104-106 are other examples of this occurrence)

Reply: Thank you very much for your constructive suggestion. We have added more references to support “Current Dilemmas in Pharmacological Treatment” section.

New added references:

[5] Sutter R, Dittrich T, Semmlack S, Rüegg S, Marsch S, Kaplan PW. Acute Systemic Complications of Convulsive Status Epilepticus-A Systematic Review. Crit Care Med. 2018 Jan;46(1):138-145. doi: 10.1097/CCM.0000000000002843. PMID: 29099419.

[6] Hawkes MA, Hocker SE. Systemic Complications Following Status Epilepticus. Curr Neurol Neurosci Rep. 2018 Feb 7;18(2):7. doi: 10.1007/s11910-018-0815-9. PMID: 29417304.

[8] Kapur J, Elm J, Chamberlain JM, Barsan W, Cloyd J, Lowenstein D, Shinnar S, Conwit R, Meinzer C, Cock H, Fountain N, Connor JT, Silbergleit R; NETT and PECARN Investigators. Randomized Trial of Three Anticonvulsant Medications for Status Epilepticus. N Engl J Med. 2019 Nov 28;381(22):2103-2113. doi: 10.1056/NEJMoa1905795. PMID: 31774955; PMCID: PMC7098487.

[13] Gaspard N, Foreman BP, Alvarez V, Cabrera Kang C, Probasco JC, Jongeling AC, Meyers E, Espinera A, Haas KF, Schmitt SE, Gerard EE, Gofton T, Kaplan PW, Lee JW, Legros B, Szaflarski JP, Westover BM, LaRoche SM, Hirsch LJ; Critical Care EEG Monitoring Research Consortium (CCEMRC). New-onset refractory status epilepticus: Etiology, clinical features, and outcome. Neurology. 2015 Nov 3;85(18):1604-13. doi: 10.1212/WNL.0000000000001940. Epub 2015 Aug 21. PMID: 26296517; PMCID: PMC4642147.

[15] Rossetti AO, Alvarez V. Update on the management of status epilepticus. Curr Opin Neurol. 2021 Apr 1;34(2):172-181. doi: 10.1097/WCO.0000000000000899. PMID: 33664203.

[16] Jehi L. Consequences of status epilepticus in the intensive care unit: what we know and what we need to know. Epilepsy Curr. 2014 Nov-Dec;14(6):337-8. doi: 10.5698/1535-7597-14.6.337. PMID: 25678866; PMCID: PMC4325589.

[17] Jehi L. Consequences of status epilepticus in the intensive care unit: what we know and what we need to know. Epilepsy Curr. 2014 Nov-Dec;14(6):337-8. doi: 10.5698/1535-7597-14.6.337. PMID: 25678866; PMCID: PMC4325589.

[20] Bateman DE. Pseudostatus epilepticus. Lancet. 1989 Nov 25;2(8674):1278-9. doi: 10.1016/s0140-6736(89)91885-0. PMID: 2573790.

[21] Appleton RE. Treatment of childhood epilepsy. Pharmacol Ther. 1995;67(3):419-31. doi: 10.1016/0163-7258(95)00023-2. PMID: 8577824.

8)  L108 – Why were these specific animal models were chosen?

Reply: Thank you for your question. Based on extensive literature research, these animal models are most widely used and accepted in the field. To make it more clear, we have added some explanations in the revised manuscript.

9)  The authors should revise techniques for reporting narrative reviews which I presume is this type of study. The method section is missing and there is no mention to how the articles were selected, which years did they choose to select articles from, which databases were searched and how they were searched, whether or not at least two independent reviewers searched for the studies, which languages the authors chose to select articles and so on. Statistical analyses should also be performed and described. This is crucial for a narrative review otherwise we do not know how to analyze the results found. I also advise the authors to display their main results in a comparative way in tables so that the readers can more easily understand the results found. I also advise authors to discuss their results taking into account previous literature.

Reply: Thank you very much for this important point. We systematically searched PubMed for publications in English with the combined keywords, including “Status Epilepticus”, “animal models”, “hypothesis”, “drug resistance”, “refractory”, “antiepileptic drugs” and “drug target”. Reference lists of relevant papers were also checked for additional studies. We have arbitrarily chosen seminal work, clinical studies with the highest level of evidence. We have also used some earlier articles and reviews, if particularly pertinent to the discussion. To make it more clear, we have added this criteria in the revised manuscript.

In addition, we have created a new table to make it easier for the reader to understand the features of different animal models of SE. The details are as follows.

Model

Mode of operation

Mechanism

Mortality rate

KA model

Intraperitoneal injection

KA binds directly to non-NMDA (KA) receptors in the neuronal postsynaptic membrane, producing excitatory postsynaptic potentials that lead to seizures.

47%-75%

Intraventricular injection

8%-21%

Intranasal injection

Lower than intraventricular injection

Pilocarpine model

lithium-pilocarpine

Pilocarpine can stimulate not only the M receptor, but also NMDA receptors, metabolic glutamate receptors, resulting in activation of the excitatory system in the brain.

27.4%-40%

Intracerebral administration

Kindling model

Hippocampus

Repeated electrical stimulations cause a gradual change in the excitatory synaptic plasticity and lower seizure threshold.

——

Amygdala

Prolonged FS

LPS-induced FS

An imbalance between the excitatory neurotransmitter glutamate and the inhibitory neurotransmitter GABA.

About 50%

Heat-induced FS.

FS induced by heat combined with LPS

Table 1. Common animal models of SE

10)  L487: Figure legends should be improved for a better understanding of the readers.

Reply: Thanks. We have changed the content to “Clinical use of medication at different times during SE attacks”

11)  L487-488: Are these figures original or they were taken from previous literature? Appropriate references should be provided.

Reply: Thanks. This figure is original and drawn by ourself. According to your suggestion. We have added corresponding references in the figure legend.

“(1) When SE=0 min, the preferred drugs are diazepam, lorazepam, clonazepam, etc. (2) When 5 min ≤ SE < 30 min, such as phenytoin, valproic acid, or levetiracetam is used as second-line medications. (3) When SE > 30 min, RSE occurs and anesthetics, including propofol, ketamine, and midazolam, can be administered.(4) when SE lasts for 24 hours, patients can choose between a ketogenic diet, hypothermia, or electroconvulsive therapy.”

12)  L439: It would be interesting if the authors could provide a schematic diagram for the new mechanisms proposed. This section should also be further expanded and authors should try to provide their own perspectives and hypothesis.

Reply: Thank you very much for this important question. We are currently more inclined to agree with the neural circuit hypothesis on this part of the mechanism. Thus, we have added some perspectives as following:

“In addition to some new molecules, we should pay more attention to the neural circuit mechanism of SE. Epilepsy is gradually accepted as a circuit-level syndrome pathologically characterized by hypersynchronous seizure activity. At present, although we have elucidated many neural circuitry mechanisms in epilepsy with the development of optogenetics, chemogenetics, in vivo imaging et al. [144,145], the neural circuit involved in different phases of SE is still largely unknown. A seizure has its own electrographic evolution feature: a seizure initiation, a spread, and a termination, which is usually linked with different networks [146]. As prolonged SE is the failure of the mechanisms responsible for seizure termination or from the initiation of mechanisms, thus revealing the circuit that medicates the termination of SE will be very important for SE treatment by using the precise intervention of the circuitry approaches.”

13)  Authors should address possible limitations of this research.

Reply: Thanks. We have added some possible limitation of this review as following:

“Meanwhile, with the advancement of technology and medical treatment, some of the data cannot be updated in real time, so we can only select the content that is currently well recognized in the field, including models and hypotheses, which is the limitation of this article.”

14) The authors should try to further elaborate a discussion and explain how do their research contributes to the field, and which conclusions did it reach?

Reply: Thank you very much for this important suggestion. Accordingly, we have added research contribution and some conclusions listed as following:

“As we summarized, there are currently many dilemmas in the pharmaco-logical treatment of SE. Despite our review of common hypotheses of drug resistance mechanisms for SE, current drugs are single mechanism, which is a daunting challenge for multicausal SE. Many new safe and effective drugs are urgently needed to be designed and developed. Meanwhile, with the advancement of technology and medical treatment, some of the lasted data cannot be updated in real time, so we can only select the content that is currently well recognized in the field, including models and hypotheses, which might be the limitation of this article. However, in this review, we provide an updated summary for the insight into drug-resistance SE, hoping to be helpful for the understanding the mechanism of refractory SE and making a rising attention to develop active compounds for both research and industry field. Here we further put forward the future perspectives for the follow-up research on SE.”

15) Overall, authors should work with English editing services to improve the text.

Reply: Thanks. An English native have helped us to polish the language in the revised manuscript.  All revisions in the revised manuscript are highlighted by red color.

Reviewer 3 Report (New Reviewer)

This is a review of mechanisms of drug-resistance in Status Epilepticus with a focus on data from animal models.

The authors present the difficulties of treating refractory status epilepticus and then proceed in presenting some types of animal models (chemical - kainate, pilocarpine, and mentions of others; electrical - kindling; and thermal). Finally, the authors present some of the proposed mechanisms.

Unfortunately, parts of the manuscript suffer from incorrect English to the level that some phrases are incomprehensible. For example, the phrase “Some studies have found that diazepam induces dose-dependently against SE 10 minutes after pilocarpine injection, and benzodiazepine was similar to diazepam, decreasing the seizure EEG in pilocarpine-treated rats.” where the meaning is completely lost.

As a review, the manuscript does not mention any methods (such as following the PRISMA guidelines, etc) or criteria on how references were selected, why they chose to focus on the particular animal models, etc.

Other than a schematic figure presenting the mechanisms of drug-resistance, there is no table or figure summarizing the data from the animal models they have presented and how these support the several hypotheses of drug-resistance mechanisms.

A minor point: since the authors also mention treatment of status epilepticus, midazolam should be mentioned with the other benzodiazepines as first line treatment.

Although this is a promising work on an important subject, language and style barriers prevent the complete scientific assessment of its quality. Major rewriting is required.

Author Response

Reviewer 3:

This is a review of mechanisms of drug-resistance in Status Epilepticus with a focus on data from animal models. The authors present the difficulties of treating refractory status epilepticus and then proceed in presenting some types of animal models (chemical - kainate, pilocarpine, and mentions of others; electrical - kindling; and thermal). Finally, the authors present some of the proposed mechanisms.

Reply: Thank you very much for your time involved in reviewing the manuscript and the valuable comments on this review. We have revised our paper according to your comments, which really improved the quality of our paper. The following are our responses in a point-by-point manner.

Unfortunately, parts of the manuscript suffer from incorrect English to the level that some phrases are incomprehensible. For example, the phrase “Some studies have found that diazepam induces dose-dependently against SE 10 minutes after pilocarpine injection, and benzodiazepine was similar to diazepam, decreasing the seizure EEG in pilocarpine-treated rats.” where the meaning is completely lost.

Reply: Thanks. An English native have helped us to polish the language in the revised manuscript.  All revisions in the revised manuscript are highlighted by red color.

As a review, the manuscript does not mention any methods (such as following the PRISMA guidelines, etc.) or criteria on how references were selected, why they chose to focus on the particular animal models, etc.

Reply: Thank you very much for this important point. We systematically searched PubMed for publications in English with the combined keywords, including “Status Epilepticus”, “animal models”, “hypothesis”, “drug resistance”, “refractory”, “antiepileptic drugs” and “drug target”. Reference lists of relevant papers were also checked for additional studies. We have arbitrarily chosen seminal work, clinical studies with the highest level of evidence. We have also used some earlier articles and reviews, if particularly pertinent to the discussion. To make it more clear, we have added this criteria in the revised manuscript.

Other than a schematic figure presenting the mechanisms of drug-resistance, there is no table or figure summarizing the data from the animal models they have presented and how these support the several hypotheses of drug-resistance mechanisms.

Reply: Thank you very much for this important point. We have created a new table to summarize this section on animal models, Please see details in the new added Table 1.

A minor point: since the authors also mention treatment of status epilepticus, midazolam should be mentioned with the other benzodiazepines as first line treatment.

Reply: Thanks. We have mentioned midazolam in the revised manuscript.

“There is also a narcotic drug, midazolam. Although many studies now find it to be more effective than diazepam, it also has stronger effects on respiration and blood pressure than diazepam, and there are limitations to its use as an anesthetic; this drug is not yet widely recognized, so we have not placed it among the first-line drugs in this article.”

Although this is a promising work on an important subject, language and style barriers prevent the complete scientific assessment of its quality. Major rewriting is required.

Reply:

Thanks again for all your constructive comments. An English native have helped us to polish the language in the revised manuscript. All revisions in the revised manuscript are highlighted by red color. We hope all above responses could address your concerns.

This manuscript is a resubmission of an earlier submission. The following is a list of the peer review reports and author responses from that submission.

Round 1

Reviewer 1 Report

This review discusses some of the mechanisms associated with drug-refractory status epilepticus. The review lacks a degree of cohesion as it encompasses several topics.  

1.       The manuscript needs extensive editing for the language. I understand that English may not be the first language for these authors, but there are professional editing services, and taking their help will improve the readability of the manuscript. I am only giving some of the examples below that would benefit from editing

Lines 65, 66: “If the patient is unable to terminate SE within this time, there is a high probability that they would evolve into refractory SE (RSE).”

Line 74, 75 “When patients enter SRSE, they can choose a ketogenic diet, hypothermia, and electroconvulsive therapy.”

Lines 81 to 83 “(1) The therapeutic window is short as mentioned before; Delayed drug treatment is easily to induce drug resistance and long-term bad consequences; (2) SE can be easily recurrent after drug treatment; (3) Error diagnosis, unable to make a clear definition.”

Lines 105, 106 “SE can last 2-6 hours, and neurons begin to damage 3 hours after KA injection.”

Lines 113, 114 “The animal will arrive at SE within 30 minutes and last for 3-12 hours.”

Lines 154 to 156 “Both KA and pilocarpine-induced SE model reflects the clinical phenomenon that SE is easily develop into resistant statue with postponed pharmacotherapeutic intervention.”

2.       While discussing the therapeutic avenues for early treatment of SE, it would be beneficial to discuss the findings of several recent clinical trials; (1) Lancet. 2020 Apr 11;395(10231):1217-1224. doi: 10.1016/S0140-6736(20)30611-5 and N Engl J Med. 2019 Nov 28;381(22):2103-2113. doi: 10.1056/NEJMoa1905795; (2) Indian Pediatr. 2020 Mar 15;57(3):222-227; Lancet. 2019 May 25;393(10186):2135-2145. doi: 10.1016/S0140-6736(19)30722-6; (3) Lancet. 2019 May 25;393(10186):2125-2134. doi: 10.1016/S0140-6736(19)30724-X.

3.       In the discussion of experimental animal models of SE, organophosphate models are not discussed. Models of neocortical injury-induced status epilepticus are also omitted. This is critical since while experimental animal studies have often targeted temporal lobe structures to induce status epilepticus, cortical insults are very common triggers of human status epilepticus that are difficult to treat. Please see these articles: (1) Epilepsia. 2020 Dec;61(12):2811-2824. doi: 10.1111/epi.16715; (2) Epilepsy Res. 1988 Mar-Apr;2(2):79-86. doi: 10.1016/0920-1211(88)90023-x.

4.       Please change this statement “After repeated bouts and massive GABA release, the synaptic membrane of GABA receptors encapsulates Cl- and internalizes it. This inactivates receptors that are no longer accessible to GABA neurotransmitters [59].” GABA-binding triggers conformational changes in the receptor leading to the opening of the ion channel and passage of chloride ions into or out of the cells depending on the concentration gradient. So the description given by these authors is incorrect.

5.       Please add a citation for this statement, lines 259-262 “Another study found that during lithium-pilocarpine-induced SE, colocalization of the β2/β3 and γ2 subunits of GABAA receptors with the presynaptic marker synaptophysin is reduced, but the proportion of these subunits is increased in dentate granule cells”

Author Response

Comments from Reviewers:

Reviewer 1

General Comment

This review discusses some of the mechanisms associated with drug-refractory status epilepticus. The review lacks a degree of cohesion as it encompasses several topics.  

Reply:

Thank you very much for your time involved in reviewing the manuscript and the valuable comments on our paper. Your comments are very helpful for revising and improving our paper, as well as the important guiding significance to our researches.

In the remainder of this letter, we discuss each of your comments individually along with our corresponding responses. To facilitate this discussion, we first retype your comments in italic font and then present our responses to the comments in blue color.

Comment 1

“The manuscript needs extensive editing for the language. I understand that English may not be the first language for these authors, but there are professional editing services, and taking their help will improve the readability of the manuscript. I am only giving some of the examples below that would benefit from editing”

Lines 65, 66: “If the patient is unable to terminate SE within this time, there is a high probability that they would evolve into refractory SE (RSE).”

Line 74, 75 “When patients enter SRSE, they can choose a ketogenic diet, hypothermia, and electroconvulsive therapy.”

Lines 81 to 83 “(1) The therapeutic window is short as mentioned before; Delayed drug treatment is easily to induce drug resistance and long-term bad consequences; (2) SE can be easily recurrent after drug treatment; (3) Error diagnosis, unable to make a clear definition.”

Lines 105, 106 “SE can last 2-6 hours, and neurons begin to damage 3 hours after KA injection.”

Lines 113, 114 “The animal will arrive at SE within 30 minutes and last for 3-12 hours.”

Lines 154 to 156 “Both KA and pilocarpine-induced SE model reflects the clinical phenomenon that SE is easily develop into resistant statue with postponed pharmacotherapeutic intervention.”

Reply 1:

Thank you very much for this comment. We have revised the manuscript carefully to improve the readability. Meanwhile, the English of our manuscript has been re-checked by an English native speaker. All the revisions are marked as red colors. The example sentences are listed as following:

Lines 65, 66 has been changed to: “If the patient fails to terminate SE within this time, there is a high probability that he/she would evolve into refractory SE (RSE), i.e., a condition that occurs when two or more ASDs including at least one non-BZDs drug are not effective.”

Line 74, 75 has been changed to: “Patients entering the stage of SRSE can choose a ketogenic diet, hypothermia, and electroconvulsive therapy.”

Lines 81 to 83 has been changed to: “(1) The therapeutic window is short as mentioned before; and delayed drug treatment can easily induce drug resistance and bad long-term consequences; (2) SE can be easily recurrent after drug treatment; (3) The disease is still exposed to incorrect diagnosis, and the lack of a clear definition.”

Lines 105, 106 has been changed to: “SE can last 2-6 hours, and neurons begin to be damaged 3 hours after KA injection.”

Lines 113, 114 has been changed to: “Usually, the animal will reach SE within 30 minutes and SE will last for 3-12 hour.”

Lines 154 to 156 has been changed to: “Both KA and pilocarpine-induced SE models reflect the clinical phenomenon that resistant status in SE can easily occur in the condition of postponed pharmacotherapeutic intervention.

Comment 2

“While discussing the therapeutic avenues for early treatment of SE, it would be beneficial to discuss the findings of several recent clinical trials; (1) Lancet. 2020 Apr 11;395(10231):1217-1224. doi: 10.1016/S0140-6736(20)30611-5 and N Engl J Med. 2019 Nov 28;381(22):2103-2113. doi: 10.1056/NEJMoa1905795; (2) Indian Pediatr. 2020 Mar 15;57(3):222-227; Lancet. 2019 May 25;393(10186):2135-2145. doi: 10.1016/S0140-6736(19)30722-6; (3) Lancet. 2019 May 25;393(10186):2125-2134. doi: 10.1016/S0140-6736(19)30724-X”

Reply 2:

Thank you very much for this constructive comment. Accordingly, we have cited these clinical trials and made some modifications in the revised manuscript as following:

Page 2 in the revised manuscript:

“In clinical studies, these three drugs, when used alone, have a nearly 50 percent chance of stopping a seizure within 60 minutes. Phenytoin is more effective than levetiracetam in convulsive status epilepticus of children, but levetiracetam may also be a good alternative for safety and patient adaptation.”

Comment 3

“In the discussion of experimental animal models of SE, organophosphate models are not discussed. Models of neocortical injury-induced status epilepticus are also omitted. This is critical since while experimental animal studies have often targeted temporal lobe structures to induce status epilepticus, cortical insults are very common triggers of human status epilepticus that are difficult to treat. Please see these articles: (1) Epilepsia. 2020 Dec;61(12):2811-2824. doi: 10.1111/epi.16715; (2) Epilepsy Res. 1988 Mar-Apr;2(2):79-86. doi: 10.1016/0920-1211(88)90023-x.”

Reply 3:

Thank you very much this valuable comment. Accordingly, we have cited these articles and added new section about “models of organophosphate neocortical injury-induced status epilepticus” in the revised manuscript as following:

Page 8 in the revised manuscript:

“In addition to the models mentioned above, there are also some other SE models. As cortical insults are very common triggers of human SE that are difficult to treat, experimental animal studies have often targeted temporal lobe structures to induce SE [61]. It has previously been reported that focal seizures can be caused by implantation of cobalt into the motor cortex [62,63]. Cobalt binds to oxygen, causing functional hypoxia, and low oxygen, the cortical tissue damage caused by trauma, hemorrhage, hypoxia, and infection often leads to acute seizures and SE [64,65]. The animals were then injected with homocysteine, an agonist for the ionized glutamate receptor for NMDA. Activation of NMDA receptors allows Ca2+ entry into neurons and enhances excitability and thus trigger prolonged seizures. However, this model can lead to edema and BBB damage, which increases mortality. Phenobarbital and lorazepam showed therapeutic effect, but no findings have been reported about drug resistance in this model [66].

Organophosphorus chemicals, including pesticide (parathion et al.) and nerve gases (diisopropyl fluorophosphate (DFP), sarin and soman et al.) can also be used to induce SE model [67]. These agents potently inhibit acetylcholinesterase, leading to the accumulation of acetylcholine and stimulating the cholinergic system, which causes to an acute cholinergic crisis. Seizures can be evident within minutes of or-ganophosphorus exposure and often progress to SE [68]. High doses of DFP were generally chosen to induce SE in laboratory animals [69-71]. DFP has a very short latency to induce SE and can reach SE within 10-20 minutes [72,73]. In this SE model, phenobarbital was found to terminate SE dose-dependently. Phenobarbital at high dose (100 mg/kg) was able to completely terminate SE, but it showed severe ad-verse effects and high mortality rates[72]”

Comment 4

“Please change this statement “After repeated bouts and massive GABA release, the synaptic membrane of GABA receptors encapsulates Cl- and internalizes it. This inactivates receptors that are no longer accessible to GABA neurotransmitters [59].” GABA-binding triggers conformational changes in the receptor leading to the opening of the ion channel and passage of chloride ions into or out of the cells depending on the concentration gradient. So the description given by these authors is incorrect.”

Reply 4:

Thanks. We have revised the description as following:

“The primary mechanism of action of benzodiazepines is to increase GABA function, but as SE continues, benzodiazepines become ineffective. It is primarily due to the internalization of synaptic (gamma-subunit) containing GABAA receptors [39,79]. During SE, both the GABAA receptor-mediated inhibition of hippocampal principal neurons and the response of these neurons to benzodiazepines are reduced [64-66]. According to Goodkin et al. report, during SE, increased internalization rates of GABAA receptors are modulated by neuronal activity, which contributed to the reduced inhibitory transmission observed during prolonged SE [67; 68]. Another study found that during pilocarpine-induced SE, colocalization of the β2/β3 and γ2 subunits of GABAA receptors with the presynaptic marker synaptophysin is reduced. In contrast, the proportion of these subunits is increased in dentate granule cells [80]. These results suggest two possible explanations for the decrease in GABAA receptors during SE. In one instance, the internalization of ligand-dependent GABAA receptors increases as the extracellular GABA concentration increases. The other is that it is believed that the increase in GABAA receptors internalization is the result of a non-ligand-independent mechanism that is mediated by the excitability of neurons, which is increased by the stimulation of excitatory amino acid receptors [81-83]. In addition, the resistance of GABAergic drugs in SE may be due to the phosphorylation status of the potassium chloride transporter KCC2, which results in the increased intracellular chloride levels, and decreased inhibition of GABAA receptor activation [84].”

Comment 5

“Please add a citation for this statement, lines 259-262 “Another study found that during lithium-pilocarpine-induced SE, colocalization of the β2/β3 and γ2 subunits of GABAA receptors with the presynaptic marker synaptophysin is reduced, but the proportion of these subunits is increased in dentate granule cells””

Reply 5:

Thanks. We have added the missing references in this statement.

New added reference:

Naylor, D.E.; Liu, H.; Wasterlain, C.G. Trafficking of GABA(A) receptors, loss of inhibition, and a mechanism for pharmacoresistance in status epilepticus. J Neurosci 2005, 25, 7724-7733, doi:10.1523/jneurosci.4944-04.2005.

Reviewer 2 Report

The manuscript “Insight into Drug-resistance in Status Epilepticus: Evidence from Animal models” by Dr. Fei Wang et al is aimed to describe the current dilemmas in pharmacotherapy of  the status epilepticus and then summaries main  advantages and disadvantages of animal models of status epilepticus. The presentation of a subject is systematic and comprehensive, list of references is quite full.

Minor criticism: Sometimes the authors use the term DZP and sometimes Diazepam. Is there any difference? if not, you need to use one term, if there is, you need to explain

I am happy to recommend the manuscript for the publication after minor corrections mentioned above.

Author Response

Comments from Reviewers:

Reviewer 2:

General Comment

The manuscript “Insight into Drug-resistance in Status Epilepticus: Evidence from Animal models” by Dr. Fei Wang et al is aimed to describe the current dilemmas in pharmacotherapy of the status epilepticus and then summaries main advantages and disadvantages of animal models of status epilepticus. The presentation of a subject is systematic and comprehensive, list of references is quite full.

Reply:

Thank you very much for your time involved in reviewing the manuscript and the very positive comments on our paper, which are very helpful for revising and improving our paper.

In the remainder of this letter, we discuss each of your comments individually along with our corresponding responses. To facilitate this discussion, we first retype your comments in italic font and then present our responses to the comments in blue color.

Comment 1

“Minor criticism: Sometimes the authors use the term DZP and sometimes Diazepam. Is there any difference? if not, you need to use one term, if there is, you need to explain.

I am happy to recommend the manuscript for the publication after minor corrections mentioned above.”

Reply 1:

Thank you very much for this comment. We have revised the format. We use diazepam uniformly.

Reviewer 3 Report

The goal of this review is to describe current dilemmas in pharmacological treatment of SE and summarize animal models used to study SE, noting the advantages and disadvantages. In addition, the authors aim to review several hypotheses involved in the mechanisms of SE.

The authors did not accomplish the objective of the review and there are several issues with this document.

1.    There are significant grammatical errors in the document along with in accurate wording and phrases. Authors should seek professional editorial guidance.

2.    Introduction: t1 and t2 is mentioned but they are not defined and reader is not directed to Figure 1.

3.    Section 2: Current dilemmas in Pharmacological… - The last sentence in the paragraph is unclear.

4.    Figure 1 – poor image quality.

5.    Organization issues. Section 2: Current dilemmas… - The first mention of dilemmas is in the 5th paragraph, this section is not well-organized. Specific dilemmas were not outlined then followed by a discussion.  Authors did not fully describe the dilemmas.

6.    Section 3 Animal models of SE – This section details the mechanism of each model, but there is very little discussion regarding the effect/impact on the pharmacology of drug resistance. There are several more detailed reviews on various SE models that goes into the advantages and disadvantages of each model, the current document does not add to published work.

7.    Section 3.1 – Paragraph 2 has numbers (1, 2,3), but what these numbers represent is not clear.

8.    Section 3.4 – FS is not defined.

9.    Section 4 – The heading “Hypothesis” is very broad, should state hypothesis of something.  

10. References missing

11. Short/incomplete sentences throughout.

12. Missing periods (.)

13.  Inconsistent abbreviations. Benzodiazepine abbreviation is used sometimes but not other times, this should be consistent.

Author Response

Comments from Reviewers:

Reviewer 3:

General Comment:

“The goal of this review is to describe current dilemmas in pharmacological treatment of SE and summarize animal models used to study SE, noting the advantages and disadvantages. In addition, the authors aim to review several hypotheses involved in the mechanisms of SE. The authors did not accomplish the objective of the review and there are several issues with this document.”

Reply:

Thank you very much for your time involved in reviewing the manuscript and the valuable comments on our paper. Your comments are very helpful for revising and improving our paper, as well as the important guiding significance to our researches.

In the remainder of this letter, we discuss each of your comments individually along with our corresponding responses. To facilitate this discussion, we first retype your comments in italic font and then present our responses to the comments in blue color.

Comment 1

“There are significant grammatical errors in the document along with in accurate wording and phrases. Authors should seek professional editorial guidance.”

Reply 1:

Thank you very much for this comment. We have revised the manuscript carefully to improve the readability. Meanwhile, the English of our manuscript has been re-checked by an English native speaker. All the revisions are marked as red colors.

Comment 2

“Introduction: t1 and t2 is mentioned but they are not defined and reader is not directed to Figure 1.”

Reply 2:

Thank you for this comment. We have added description about the definition of t1 and t2 as following:

Page 1 in the revised manuscript:

“These two time points has clear clinical implications: The t1 determines the time at which treatment should be considered or started, whereas the t2 determines how aggressively treatment should be implemented to prevent long-term consequences.”

Comment 3

“Section 2: Current dilemmas in Pharmacological… - The last sentence in the paragraph is unclear.”

Reply 3:

Thanks. Accordingly, we have revised corresponding sentence as following:

Page 2 in the revised manuscript:

“At present, although there are many ASDs available to be chosen, the following dilemmas are still facing great challenges for contemporary drug treatment: (1) The therapeutic window is short as mentioned before; and delayed drug treatment can easily induce drug resistance and long-term bad consequences; (2) SE can be easily recurrent after drug treatment; In recent years, studies have found that the recurrence rate of SE in adult patients is ~20% in the next four years [14,15]. (3) SE is still exposed to incorrect diagnosis. Non-epileptic seizures can be misdiagnosed with genuine epileptic seizures, which may lead to inappropriate, costly and potentially harmful treatment. SE is a life-threatening disorder, which makes the early recognition necessarily important.

Then, what makes this situation and how to address these dilemmas? On one hand, from the source of the current ASD, most of them are screened through acute animal models [16]. In fact, as epilepsy is a chronic disease, there is a great lack of screening candidate compounds in drug-resistant chronic epilepsy models or drug-resistant SE models. On the other hand, the most of the current ASD produce anti-seizure actions with the mechanism of "excitation-inhibition imbalance", and lack of new drug targets and new mechanisms to start with so, it is very important to study mechanism of refractory SE and develop effective drugs.”

Comment 4

“Figure 1 – poor image quality.”

Reply 4:

Thanks. We have uploaded all high-resolution figures during this revision.

Comment 5

“Organization issues. Section 2: Current dilemmas… - The first mention of dilemmas is in the 5th paragraph, this section is not well-organized. Specific dilemmas were not outlined then followed by a discussion. Authors did not fully describe the dilemmas.

Reply 5:

Thank you very much for this constructive comment. Accordingly, we have re-organized the section about “current dilemmas” part. Notably, we have outlined specific dilemmas and added more detailed discussion. Please see details in Section 2.

Comment 6

“Section 3 Animal models of SE – This section details the mechanism of each model, but there is very little discussion regarding the effect/impact on the pharmacology of drug resistance. There are several more detailed reviews on various SE models that goes into the advantages and disadvantages of each model, the current document does not add to published work.”

Reply 6:

Thank you very much for this constructive comment. We agree with you that previously there are several more detailed reviews on various SE models that goes into the advantages and disadvantages of each model. Our review focus on the termination of SE. According to your comment, to distinguish the difference with previous reviews, we have added more detailed discussion about the drug resistance feature for SE termination in these SE models. Pleases see details in the Section 3. All the revisions are highlighted by red color.

Comment 7

“Section 3.1 – Paragraph 2 has numbers (1, 2, 3), but what these numbers represent is not clear.”

Reply 7:

Thanks. In the original version of manuscript, numbers (1,2,3) represent different types of KA delivery to induce SE model. To avoid any confusion, we have deleted these numbers and revised the corresponding descriptions in the revised manuscript.

Comment 8

“Section 3.4 – FS is not defined.”

Reply 8:

Thanks. FS is short for “Febrile seizure”. We have added this definition in the revised manuscript.

Page 2 in the revised manuscript:

“The main currently-used SE models currently used according to the characteristics of SE that has long-lasting or repeated multiple seizures, including kainic acid (KA), pilocarpine, kindling, and prolonged febrile seizure (FS) models”

Comment 9

“Section 4 – The heading “Hypothesis” is very broad, should state hypothesis of something.”

Reply 9:

Thank you very much for this suggestion. We have added detailed description about each hypothesis.

Comment 10

“References missing”

Reply 10:

Thanks. We have re-check all contents and added the missing references in the revised manuscript.

Comment 11

“Short/incomplete sentences throughout.”

Reply 11:

Thanks. We have revised the manuscript carefully to improve the readability. Meanwhile, the English of our manuscript has been re-checked by an English native speaker.

Comment 12

“Missing periods (.)”

Reply 12:

Thank you very much for your detailed review. We have revised this typo throughout the manuscript.

Comment 13

“Inconsistent abbreviations. Benzodiazepine abbreviation is used sometimes but not other times, this should be consistent.”

Reply 13:

Thanks. We have re-checked and unified the format throughout the manuscript.

Round 2

Reviewer 3 Report

As stated in my initial review, the document has significant grammatical errors and the authors should seek professional service to correct these errors.

The authors stated that “the English of our manuscript has been re-checked by an English native speaker.” This is not sufficient because the document still has significant grammatical errors and unsuitable phrasing. Starting from the abstract, there are still errors from the first submission and new errors were added during the revision.